# Variations and Patterns in Sleep: A Feasibility Study of Young Carers in Families with ALS

**DOI:** 10.3390/jcm10194482

**Published:** 2021-09-29

**Authors:** Melinda S. Kavanaugh, Kayla T. Johnson, Matthew J. Zawadzki

**Affiliations:** 1Helen Bader School of Social Welfare, University of Wisconsin-Milwaukee, 2400 E Hartford Avenue, Milwaukee, WI 53211, USA; 2Department of Psychology, University of Wisconsin-Milwaukee, 2400 E Hartford Avenue, Milwaukee, WI 53211, USA; kleinra2@uwm.edu; 3Psychological Sciences, University of California Merced, 5200 N Lake Road, Merced, CA 95343, USA; mzawadzki@ucmerced.edu

**Keywords:** young carers, caregiving, sleep

## Abstract

Introduction: Children and youth under the age of 19 provide daily care for family members living with illness, including Amyotrophic Lateral Sclerosis (ALS). Caregiving affects school performance, social support, stress, and anxiety. Yet, little is known about potential disruptions in sleep. Methods: A quasi-experimental matched comparison of age- and gender-matched young carers (*n* = 8) and non-carers (*n* = 12) was used in this study. Participants completed a pre/post survey, wore an actigraphy device, and journaled sleep/wake times for 5 days. Results: Young carers had shorter sleep duration (*t* = 51.19 (11.99)), efficiency (*t* = 55.49 (14.00)), sleep quality (*t* = 51.32 (12.26)), and higher rates of utilizing sleep medications (*t* = 50.81 (11.49)). The case study sleep data showed that carers had lower total sleep time (CG = 6.75 ± 1.47, NCG = 7.08 ± 1.36) and sleep efficiency than non-caregivers (0.80 ± 0.23). Case examples were reported across groups. Conclusions: The study results demonstrate feasibility, while providing crucial initial case data on sleep quality in young carers. The findings underscore the need to better document the impact of caregiving on young carer’s well-being across several areas, including sleep. This data has implications for larger scale studies examining how sleep disruption impacts well-being more broadly and in providing support and respite interventions for young carers across disorders.

## 1. Background

Children and youth under the age of 19, “young carers”, are actively involved in caregiving including bathing, feeding, toileting, and managing medications and complex assistive devices [1,2]. Despite young carers representing almost 10% of the caregiving population in the U.S. [3], few care programs and services target them [4]. Young carers experience depression and anxiety [5], and have little support from non-caregiving youth [4]. Care tasks impact school, with youth often forgoing school and peer activities in order to care [6]. Young carers participate in emotionally difficult and demanding care well into the evening, including for complex and difficult to manage diseases such as amyotrophic lateral sclerosis (ALS) [7]. In the U.S., young carers in families with ALS engage in care for up to 5 h a day [8], including into the evening hours, potentially affecting sleep.

Given the potential for late-night and early-morning care tasks, sleep becomes a critical construct to examine for young carers. Youth, in general, need more sleep than adults, ranging from 9–11 h for school-aged children and 8–10 h for teenagers, compared to just 7–9 h for adults [9]. Poor and disrupted sleep in youth is related to poor academic performance [10], and negative physical and mental health consequences [11], all of which may be exacerbated in young carers whose routines and sleep may be disrupted with care tasks, anxiety, and worry about the care recipient. Moreover, young carers are at a critical developmental period for setting positive health habits, and adolescents who sleep less than 8 h a day display early risk factors for later life obesity, higher body mass index, more body fat, and more sedentary activities [12].

Although qualitative research has described lack of sleep in young carers [13], objective assessments of sleep health in young carers are missing from the data. This follows a general trend in the science of young carers that is predominately descriptive and reliant on self-report, including measures of health and well-being [14]. As such, no known objective sleep data exists for young carers, including whether they are compliant with wearing an actigraphy device, as has been tested in non-caregiving youth [12,15,16]. Conducting a sleep assessment through a non-invasive, and easy to use wearable actigraphy device is critical given young carers’ struggle to complete normal childhood activities such as schoolwork [17,18] and peer activities [6], while often maintaining a full caregiving load.

With no known published data on sleep disruption in young carers and the need to establish feasibility in a taxed and vulnerable population, this project describes the initial feasibility of collecting 24-h patterns of sleep in young carers living in families with ALS.

The aims of the study were:(1)To establish the feasibility of recruitment, retention, and compliance in young carers and a non-caregiving comparison group.(2)To use case examples to assess differences in sleep variation and patterns between young carers and non-caregiving controls.

## 2. Methods

### 2.1. Study Design

This quasi-experimental matched comparison study utilized a wearable GENEActiv device (Activinsights Ltd., Kimbolton, UK) to collect objective sleep data along with electronic surveys and paper/pencil sleep journals for 5 consecutive 24-h periods. The study was approved by the principal investigator’s (PI) Institutional Review Board (IRB). Participants under 18 gave signed assent, while parents provided consent. Participants aged 18 and over signed consent.

### 2.2. Recruitment

The project was a partnership with an ALS Multidisciplinary Clinic at a large midwestern teaching hospital, the state chapter of the ALS Association, and the PI’s university. Persons living with ALS and their young carers were recruited from both the clinic and ALS chapter. The ALS chapter staff received study information to send to families via email, newsletters, and support groups. The study’s PI attended the weekly ALS multidisciplinary clinic to meet with families and provide study information. Gender and age matched non-caregiving youth were recruited via word of mouth via the research team and PI university faculty colleagues. The latter was particularly important to facilitate the matching process for non-caregiving participants. All study materials detailed the number and email of the study PI, and instructed interested families to call the PI for more information.

### 2.3. Participants

The inclusion criteria for young carers included the following: (1) between the ages of 10–19; (2) have a family member living with ALS; (3) participate in some measure of care for a family member; (4) fluent in English; (5) no history of obstructive sleep apnea to avoid confounds related to assessment of sleep quality (as confirmed by parents during screening); and (6) not pregnant. Non-caregiving youth comparison group participants were identified and specifically matched to the young carers by age and gender, while meeting the following criteria: (1) report not providing care for an ill family member; (2) fluent in English; (3) no history of obstructive sleep apnea (as confirmed by parents during screening); and (4) not pregnant.

### 2.4. Data Collection

After screening by the study team for inclusion, the initial study visit took place either at the participant’s home or at another location of the participant’s choosing. After obtaining consent/assent, the youth had their height and weight measured by study staff. Participants were then given a three-digit ID and were asked to complete the baseline survey on a study tablet provided by study staff. Upon completion of the survey, participants were then given a sleep journal and instructions on how to fill out their sleep journals for the next 5 days. They were instructed to note any care tasks (for caregivers), and sleep and wake times for each of the 5 days as close to those events as possible. Participants were then given a GENEActiv device (Activinsights Ltd., Kimbolton, UK) [19] and trained how to correctly wear and adjust it to ensure attainment of a strong signal. The GENEActiv device has been validated in several studies for assessing sleep in youth [20,21]. Prior work with the GENEActiv device with youth has found that 3–5 days was sufficient to produce reliable estimates of sleep [22]. We opted for the upper range to ensure enough days of data collection but did not expand this window to avoid potentially overburdening a chronically stressed population. Youth were provided written and pictorial instructions for the device to ensure they wore the device at all times and did not take it off until the end of the 5-day period. Exceptions allowed for removal of the device (with immediately putting it back on) if swimming or other immersive water activity.

A follow-up visit was scheduled approximately 6 days after the initial visit, either at the participant’s home or location of their choosing. During the follow-up visit, participants returned all study materials and completed a post survey on a study tablet. They received a gift card for each day they completed the study procedures (wearing the device, completing the journal). Alternatively, if participants lived over an hour away, they were given a pre-paid USPS box addressed to the PI for returning the watches and journals. They were given the post survey via an email link, and the gift cards were mailed to them when the materials were returned.

### 2.5. Measures

#### 2.5.1. Demographics

Participants reported their age in years, education level (0 = elementary, 1 = middle, 2 = high school, 3 = college), ethnicity (white, non-white Hispanic, Black/African American, Asian, Pacific Islander, other) and gender using an open-ended format allowing for personal definition.

#### 2.5.2. Caregiving

The Multidimensional Assessment of Caring Activities Checklist (MACA-YC18) [14] was utilized to assess baseline caregiving duties and how often each was performed in the past month. The list includes 18 caregiving tasks from watching over to bathing, feeding, and toileting. Participants answered how often they complete each task on a scale ranging from 0 = Never to 1 = At least some of the time.

#### 2.5.3. Sleep

Sleep quality was assessed using the Pittsburgh Sleep Quality Index (PSQI) [23], a 22-item measure assessing seven dimensions of sleep: sleep duration, sleep disturbance, sleep latency, day dysfunction due to sleepiness, subjective sleep quality, medication usage, and sleep efficiency. The PSQI has been used extensively in youth populations, with research showing, for example, that it produces reliable and valid estimates of sleep quality [24], and is sensitive enough to detect change in sleep quality in response to interventions [25]. Items for each dimension are combined and rescored to range from 0 to 3; then scores across all seven dimensions are combined to produce a possible range of 0 to 21. The PSQI total score is a validated predictor of several sleep disorders in at-risk persons [23,26].

#### 2.5.4. Sleep Journal Measures

Self-reported bedtimes and awake times were used to assess total duration of sleep and sleep latency (time in bed before sleep).

#### 2.5.5. GENEActiv

Validated GENEActiv devices were used to measure sleep duration and efficiency via activity [27]. To facilitate analyses, each participant’s data were converted into 1-min epochs using the associated GENEActiv PC processing software (v.3.3). Each data file was then run through an open source macro developed by the manufacturer (https://www.activinsights.com/; accessed on 10 March 2020) in order to calculate wear time (number of hours the GENEActiv device was on the participant’s wrist), total time in bed, total sleep time, and sleep efficiency using the defined algorithm from the manufacturer. As part of the data cleaning, the following rules were applied: (1) GENEActiv sleep measures were used after they were confirmed with self-reported sleep times (when diary sleep times were within one hour of watch sleep times); (2) daytime data incorrectly marked as sleep was corrected based on journal data (e.g., if a participant wrote they took the watch off for a sport and the watch data counted that as sleep); (3) we removed data that was counted as sleep prior to self-reported sleep if the watch showed activity until their self-reported sleep; and (4) we used the sleep times from the GENEActiv watch if none of the above rules applied (i.e., if the activity on the watch did not dispute their self-reported sleep time), or if the cleaning process disrupted the macro file to where it clearly incorrectly calculated sleep and wake times.

#### 2.5.6. Role of the Funding Source

This study was funded by the Clinical and Translational Science Institute (CTIS) of Southeastern Wisconsin. While CTSI provided the funds for the project, they had no other engagement in the research or publication of this paper. All authors had full access to the data and accept responsibility for the publication of this paper.

## 3. Results

### 3.1. Participants

As shown in Table 1, the participants (caregivers = 8, non-caregiving controls = 12; *n* = 19 for actigraphy data, *n* = 20 for survey data) ranged in age from 9 to 19 years old. The average age of the participants was 14. Participants identified as mostly female (*n* = 11) and male (*n* = 8) with one participant preferring not to answer. The majority of participants identified as white.

### 3.2. Caregiving

Care tasks are shown in Table 2; young carers assisted with multiple care tasks including household chores (*n* = 68, 100%), walking (*n* = 6; 75%), feeding (*n* = 4; 50%), transferring (*n* = 4; 50%), dressing (*n* = 3; 38%), and helping with range of motion exercises (*n* = 4; 50%).

### 3.3. Feasibility

Feasibility was established through the recruitment of both young carers and non-caregiving controls. Participants wore the device on average 23.17 h a day (99.2% of the study time), with no significant difference between the caregiving (23.69 ± 1.37) and non-caregiving sample (23.86 ± 0.35), *t*(72) = 0.813, *p* > 0.05. The times at which the watch appeared inactive during the day corresponded to participants’ journals reporting their engagement in self-hygiene (e.g., showers) or high physical activity (e.g., volleyball, basketball), indicating strong compliance with the protocol of when to wear the watch and noting any non-wear moments. All youth completed the sleep portion of the journals, while the caregivers completed the additional caregiving portion of the journals. Moreover, there were no reported adverse events from wearing the watches or completing the journals.

### 3.4. Self-Reported Sleep Case Study Data

Table 3 details the self-reported sleep dimensions from the PSQI and describes individual sleep and an overall sleep quality score. Caregivers reported shorter sleep duration (*t* = 51.19 (*p* = 0.70)), latency (*t* = 52.42 (*p* = 0.39)), efficiency (*t* = 55.49 (*p* = 0.06)), poorer overall sleep quality (*t* = 51.32 (*p* = 0.67)), and higher rates of utilizing medications to help them sleep (*t* = 50.81 (*p* = 0.77)) than non-caregivers. However, these t- scores are not significantly different between groups.

### 3.5. Objective Case Study Sleep Data

The GENEActive device showed caregivers (10.67 ± 4.4) spent more time in bed than non-caregivers (9.58 ± 3.72), but had lower total sleep time (CG = 6.75 ± 1.47, NCG = 7.08 ± 1.36). Moreover, caregivers (0.70 ± 0.23) had lower sleep efficiency than non-caregivers (0.80 ± 0.23). Non-caregivers displayed a more typical sleep pattern without waking, restlessness, and other sleep disturbances, yet did not all uniformly sleep the “typical” length of time for youth (i.e., 8.11 h).

### 3.6. Case Examples

Full participant sleep data is shown in Table 4. However, to help illustrate some of the patterns and variations in sleep observed and recorded in non-caregivers, but also across young carers, we descriptively present four nights of data.

Case 1 non-caregiver: The first case, as shown in Figure 1a, is from a non-caregiver and represents a relatively expected and healthy night’s sleep. The participant went to bed at 9:20 p.m. and woke up at 7:06 a.m., with a sleep duration of 8.4 h. There was no activity shown on the actigraphy device during this sleep window, indicating little, if any, restlessness (as shown by the bottom line in Figure 1a).

Case 2 young carer: As shown in Figure 1b, in case 2, the participant went to bed at 2:00 p.m. and woke at 11:00 a.m., with a sleep duration of 8.7 h. The odd hours are reflected in the journal and the care needs of her parent—the participant provided care when the well-parent worked overnight and third shifts. During sleep hours, the activity on the device shows a disrupted caregiver sleep pattern, with little activity during the non-sleep hours. These data were corroborated by her journal entries, which showed she got up for care tasks in the night, and did less activity during the day due to naps to make up for the sleep disruption.

Case 3 young carer: As shown in Figure 1c, the young carer in case 3 went to bed at 9:35 p.m. and woke at 5:57 a.m., with a sleep duration of 5.1 h. The activity data details the clearly fragmented sleep pattern of the caregiver, showing that they woke frequently in the night. However, the participant did not note any wakefulness in their journal, despite the device clearly detailing the waking.

Case 4 young carer: As shown in Figure 1d, case 4 shows short sleep duration in a caregiver. The participant went to bed at 11:08 p.m. and woke up at 6:40 a.m. While the elapsed sleep period was 7.5 h, their sleep time was 4.9 h, indicating that they spent a significant amount of time in bed awake.

## 4. Discussion

This study established the feasibility of assessing sleep duration and quality in a taxed and vulnerable population, young carers, while providing initial descriptions of the differences between these young carers and a comparison group of non-caregivers. All participants, both caregivers and non-caregivers, wore the GENEActiv devices, and completed their journals and online surveys during the 5-day period. No dropouts were reported in either group. Thus, the initial goal of establishing feasibility was reached. Moreover, the use of journaling and GENEActiv devices provided an opportunity to assess sleep from both a subjective and an objective perspective, which are critical in showing the accuracy of youth in detailing sleep/wake patterns, timings, and activities in a vulnerable youth population.

Young carer data highlights the complexity and duration of care engaged in by youth in the family, yet there are limited data addressing the potential health implications of care, including sleep health. Previous research found that youth worry not just about the health of the person for whom they provide care, but also for their own health [28], highlighting the need to identify and target the health of young carers. As reflected in previous data on young carers in ALS, young carer participants in this study were very involved in care, with activities ranging from feeding to watching and assisting with transferring. While not all young carers detailed care in the evening or at nighttime, all caregiving participants engaged in care at points throughout the day/night, which may be impacting their sleep either literally through activity, or subconsciously through worry and anxiety throughout the night.

Caregiver participants reported shorter sleep duration, latency, efficiency, and poorer overall sleep quality than non-caregivers, findings that are supported in qualitative explorations of young carers, and anecdotal evidence from clinical practice. However, an unexpected finding was the reported higher usage of medication by young carers compared to the non-caregiving group. While the findings of more disrupted sleep are not unexpected given the evidence of sleep disruption in adult caregivers [29], the use of medications for sleep in young carers is concerning. Young carers may have access to medications not available to other youth, given their care role. Thus, they may see medications as accessible in ways that non-caregiving youth do not, and furthermore, see medications as a way to deal with care stress. Indeed, previous data highlights the emotional toll [5] and social isolation [4] of being a young carer, potentially influencing their use of medications for sleep or to simply manage being a caregiver.

Caregivers spent more time in bed than non-caregivers, had less total sleep time, and subsequently lower sleep efficiency—all critical for healthy development. In youth, lack of sleep or disrupted sleep can lead to severe daytime dysfunction [30,31], and has clear implications for long term health effects including heart disease and obesity [32].

Yet, how consistent caregiving may exacerbate sleep disruption is unknown. The youth in this small study were engaged in a variety of care tasks, many which may intensify as more symptoms appear or worsen. While the trajectory of ALS may be brief, lasting typically 2–5 years, in other illnesses, care may cover the duration of childhood in young carers. Thus, assessing the length and intensity of care regardless of disease is critical to help youth develop healthy sleep patterns over time. Indeed, addressing time spent providing care may have far reaching impacts beyond care for the persons with ALS, and may serve to reduce the potential long-term impacts of sleep disruption due to caregiving, worry, and anxiety of care.

While both non-caregiving and caregiving youth completed the sleep journal, for the caregiving youth, several inconsistent entries were found (e.g., not writing caregiving tasks down when they reported that they engage in daily care tasks in the surveys and recruitment process). While these omissions may have been due to youth being preoccupied and forgetting given the stress in their lives, it may also reflect the lack of “labeling” themselves as caregivers. Young carers do not always identify the tasks they engage in as “caregiving” or see themselves as the caregiver, rather they engage in tasks as part of “normal” life. In the baseline survey the same youth described care tasks, yet in the moment, they may not be perceived as such. Adult caregiving research highlights the “care persona” taken by adult caregivers, yet how this applies to youth is unclear, including how the perception of care influences their perception of self.

Finally, young carers may not be waking to provide care, yet they still experience disrupted sleep, even when they do not know sleep is being disrupted, as shown in case 3. This disruption may be a manifestation of the stress, anxiety, and depression experienced by many young carers [5], and underscores the need to assess the potential impact of anxiety or worry, and develop stress reduction interventions for young carers. Yet, sleep is not the only disruption found in young carers. Data from earlier work on ALS, found youth are often too worried to concentrate in class [8], underscoring the potential for worry and anxiety to not only impact sleep, but to also affect other farther-reaching activities including school performance and attendance, leading to the need for intervention. Indeed, evidence from adult caregivers’ mindfulness interventions showed improvement in PSQI in adult caregivers [33], which underscores the potential utility in reducing PSQI scores and improving sleep health in young carers through mindfulness and relaxation interventions.

### Limitations

While the use of GENEActiv devices had several positive attributes, there were limitations. First, the difference in scores, but the lack of statistical significance, may be due to the small sample size, and limits the ability to generalize beyond the sample discussed. Second, we used the PSQI as a measure of sleep quality. However, future research may wish to assess additional dimensions of sleep, particularly levels of daytime fatigue and sleepiness. Third, we relied on parental reporting of obstructive sleep apnea—although these reports have been shown to be valid in detecting actual levels of sleep apnea [34], we may have missed some cases by not measuring in the youth directly. Finally, this study was limited to young carers in families with ALS, and were known to ALS organizations and clinics, limiting the generalizability to all young caregiving groups, and limiting our knowledge of how these youth may differ from youth not engaged with the ALS clinic or Association. Future research should address health broadly to include other health measures in addition to sleep, including hormone stress measures in young carers. However, given the paucity of research on sleep in the young carer population overall, it is hoped that the results will be used as a beginning foray into understanding how sleep patterns vary across young carer groups and disease states, and for assessing the best ways to measure sleep in an isolated and underserved population and how to develop interventions.

## 5. Conclusions

Young carers and their non-caregiving counterparts will use an actigraphy device and journal to capture sleep activity, thus laying the groundwork for larger studies of young carers across diseases and care needs. Clear avenues exist for future research to assess sleep in larger samples, but also to understand differences and variations in the care experience of the young carer themselves, including how care tasks relate to their own well-being. Finally, future work must address the accessibility of medications, for sleep or otherwise, in this vulnerable and understudied youth population.

## Figures and Tables

**Figure 1 jcm-10-04482-f001:**
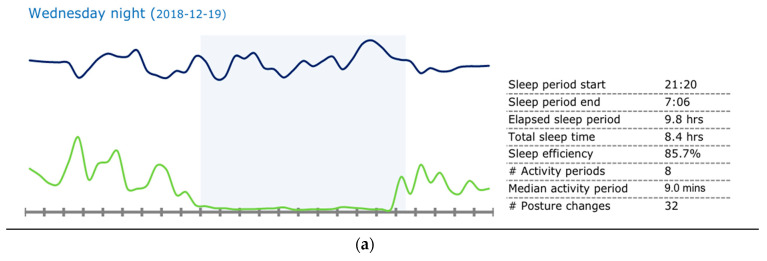
Case studies of sleep patterns. Note: GENEActiv graphs depicting various sleep patterns. The top line indicates temperature, the bottom line indicates activity, and the shaded area indicates sleep. This graph illustrates the period from noon to noon the next day. (**a**) Participant 103 (non-caregiver) showing a typical, ideal sleep pattern. (**b**) Participant 107 (caregiver) showing daytime rest. (**c**) Participant 114 (caregiver) showing sleep fragmentation (i.e., frequent night waking). (**d**) Participant 101 (caregiver) showing short sleep duration.

**Table 1 jcm-10-04482-t001:** Descriptive statistics.

	CG	NCG
**Age**		
Average age	13	16
Age range	9–19	9–17
**Gender**		
Male	*n* = 3	*n* = 5
Female	*n* = 5	*n* = 6
No answer	*n* = 0	*n* = 1
**Race**		
White/Caucasian	*n* = 8	*n* = 11
Black/African American	*n* = 0	*n* = 1
Asian	*n* = 0	*n* = 2
Native American	*n* = 0	*n* = 1
Hispanic		*n* = 1
Other	*n* = 0	*n* = 0
No answer	*n* = 0	*n* = 1
**Education level**		
Elementary School	*n* = 1	*n* = 2
Middle School	*n* = 4	*n* = 4
High School	*n* = 2	*n* = 5
College	*n* = 0	*n* = 0
Not currently in school	*n* = 1	*n* = 0
No answer	*n* = 0	*n* = 1

Note: CG = caregiver, NCG = non-caregiver. For ethnicity, participants were instructed to pick all that applied, thus some had more than one selected.

**Table 2 jcm-10-04482-t002:** Care tasks.

Tasks	At Least Some of the Time (%)	Never (%)
Household chores	8 (100)	0 (0)
Grocery shopping	6 (75)	2 (25)
Cook meals	4 (50)	4 (50)
Assist with dressing	3 (38)	5 (63)
Assist with bathing	1 (13)	7 (88)
Assist with toileting	2 (25)	6 (75)
Transferring	4 (50)	4 (50)
Assist with walking	6 (75)	2 (25)
Assist with feeding	4 (50)	4 (50)
Interpret or use communication device	3 (38)	5 (63)
Administer medication	2 (25)	6 (75)
Assist with respiratory equipment	2 (25)	6 (75)
Clean mouth/drool	3 (38)	5 (63)
Keep company	7 (88)	1 (13)
Take care of siblings		
Assist with writing	2 (25)	6 (75)
Help with range of motion exercises	4 (50)	4 (50)

**Table 3 jcm-10-04482-t003:** Mean t-scores (SD) for caregivers (CG) and non-caregivers (Non-CG) on the PSQI survey.

	CG	NCG	*p* Value
PSQIDURAT	51.19 (11.99)	49.30 (9.16)	0.703
PSQIDISTB	54.60 (15.20)	46.93 (0.00)	0.094
PSQIDAYDYS	47.16 (7.69)	51.89 (11.20)	0.312
PSQISLPQUAL	50.00 (11.07)	50.00 (9.73)	1.0
PSQIMEDS	50.81 (11.49)	49.46 (9.38)	0.776
PSQILATEN	52.42 (10.74)	48.39 (9.60)	0.391
PSQIHSE	55.49 (14.00)	46.80 (5.14)	0.066
PSQI	51.32 (12.26)	49.23 (8.94)	0.674

Note. CG = caregiver, NCG = non-caregiver, higher PSQI scores indicate worse sleep.

**Table 4 jcm-10-04482-t004:** Participant log.

ID	Age	Gender	Caregiver Status	Sleep Duration	Sleep Fragmentation Present (Y/N)	Daytime Inactivity (Y/N)	Daytime Care Noted (Y/N)	Nighttime Care Noted (Y/N)	Percent Time Watch Worn
101	10	F	1	6.58	N	N	N	N	100
102	13	F	0	8.75	N	N	n/a	n/a	100
103	10	F	0	7.73	N	N	n/a	n/a	98.7
104	10	F	1	7.13	N	N	N	N	100
105	13	F	1	n/a	n/a	n/a	N	N	n/a
106	15	M	1	6.25	Y	Y	Y	N	91.3
107	19	F	1	7.39	Y	Y	N	Y	100
108	12	F	1	6.10	N	N	N	N	100
109	16	M	1	7.58	Y	N	N	N	100
110	10	F	0	5.87	Y	N	n/a	n/a	97.4
111	17	F	0	7.34	N	Y	n/a	n/a	100
112	12	F	0	7.36	N	N	n/a	n/a	100
113	15	M	0	7.87	Y	N	n/a	n/a	99.8
114			1	6.21	Y	N	N	N	100
115	10	M	0	8.00	Y	Y	n/a	n/a	100
116	14	F	0	6.22	Y	N	n/a	n/a	97.9
117	15	M	0	7.23	N	Y	n/a	n/a	100
118	17	M	0	6.93	N	Y	n/a	n/a	100
119	16	M	0	5.47	N	N	n/a	n/a	99.4
120	12	M	0	6.16	N	Y	n/a	n/a	100

Note: Caregiver status 0 = non-caregiver and 1 = caregiver. Y = yes and N = no. Participant 105′s watch failed to record activity.

## Data Availability

Any anonymized data will be shared by request from the principal investigator.

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
