# Peer review of "Variations and Patterns in Sleep: A Feasibility Study of Young Carers in Families with ALS"

_jcm, 2021, doi:10.3390/jcm10194482_

Round 1
Reviewer 1 Report
This is a quasi-experimental case-control study investigating the impact of daily care burden on sleep in young carers in family of amyotrophic lateral sclerosis (ALS). The results showed young carers had shorter sleep duration, efficiency, sleep quality and higher rates of taking sleep medications by Pittsburgh Sleep Quality Index (PSQI). By the say, young carers had lower total sleep time and sleep efficiency than non-caregivers in actigraphy measured by "GENEActiv" accelerometery. These findings are expected. Four major issues are needed to be addressed:
- The rationale of small sample size based on power calculation and statistical analysis in terms of parametric vs. non-parametric methods should be given.
- For the age between 10-19, the only inclusion criteria for sleep disorders "no history of obstructive sleep apnea (OSA)" is not enough. First of all, there was lack of both subjective and objective evidence for OSA. The second, other possible sleep disorders for young people such as insomnia, restless legs syndrome with periodic limb movement in sleep and circadian rhythm disorders should also be excluded.
- Only PSQI was used for sleep-related measures. Sleep scales for excessive daytime sleepiness such as Pediatric Daytime Sleepiness Scale and scales regarding health-related quality of life, depression and anxiety and fatigability might be needed.
- The data collected by "GENEActiv" for only five days are also not enough. Data collected for at least 14 days across two weekends can reflect a more objective sleep-wake cycle and sleep efficiency.
Thus the results should be biased based on the above issues.
Author Response
Please see attached response to reviewers
Reviewer 2 Report
Although the authors analyzed a sleep pattern between young carers and non-caregiving controls, there is little concern that the experimental schedule is inappropriate. Thus, I listed some points that need to be clarified and correct below.
[Major point]
- Although the authors performed a comparative analysis between 13 CG and 16 NCG, the size of the cohort is more “case report” rather than “research paper.”
- According to the previous report (10.1186/s12966-021-01143-6), GENEActiv devices need 3 to 5 nights and days for reliable estimates. The experimental schedule of the present study had no time for acclimatization to the device. The authors should reconsider the experimental schedule and remeasure the data again.
[Minor point]
- Table 1 and Table 2 are immature and need to be reconstructed into a format similar to the table in the following paper; 10.1503/cmaj.1031730, 10.1017/S0950268814003665.
- In line 210, there was a typo; “GENEActive.” The authors should correct.
Author Response

(The authors gave the same response as above.)

Round 2
Reviewer 1 Report
- First of all, I don't think the results of PSQI coming from a case- control study with a small sample size are persuading. Sample size calculation to avoid type II statistical error is mandatory. Normality test is also needed to support using parametric Student's t test. Median and IQR of the sub-scales of PSQI and non-parametric Mann-Whitney U test might be better.
- The second, there was lack of statistical analysis to compare the objective data obtained from actigraphy. Thus the results showed caregivers (10.67 ±4.4) spent more time in bed than non-caregivers (9.58 ±3.72), but had lower total sleep time (CG = 6.75 ± 1.47, NCG = 7.08 ± 1.36). Moreover, caregivers (.70 ± .23) had a lower sleep efficiency than non-caregivers (.80 ± .23) should be biased.
Reviewer 2 Report
The manuscript has been much improved and is in a nice condition now.